# Distancing the socially distanced: Racial/ethnic composition's association with physical distancing in response to COVID-19 in the U.S.

**Joseph Gibbons** *

Sociology, San Diego State University, San Diego, California, United States of America

* jgibbons@sdsu.edu

**Data Availability Statement:** American Community Survey data can be access through the Census API or though the website American Factfinder (https://data.census.gov/cedsci/). Google Mobility data can be accessed through a

## Abstract

Social distancing prescribed by policy makers in response to COVID-19 raises important questions as to how effectively people of color can distance. Due to inequalities from residential segregation, Hispanic and Black populations have challenges in meeting health expectations. However, segregated neighborhoods also support the formation of social bonds that relate to healthy behaviors. We evaluate the question of non-White distancing using social mobility data from Google on three sites: workplaces, grocery stores, and recreational locations. Employing hierarchical linear modeling and geographically weighted regression, we find the relation of race/ethnicity to COVID-19 distancing is varied across the United States. The HLM models show that compared to Black populations, Hispanic populations overall more effectively distance from recreation sites and grocery stores: each point increase in percent Hispanic was related to residents being 0.092 percent less likely (p< 0.05) to visit recreational sites and 0.127 percent less likely (p< 0.01) to visit grocery stores since the onset of COVID-19. However, the GWR models show there are places where the percent Black is locally related to recreation distancing while percent Hispanic is not. Further, these models show the association of percent Black to recreation and grocery distancing can be locally as strong as 1.057 percent (p< 0.05) and 0.989 percent (p< 0.05), respectively. Next, the HLM models identified that Black/White residential isolation was related to less distancing, with each point of isolation residents were 11.476 percent more likely (p< 0.01) to go to recreational sites and 7.493 percent more likely (p< 0.05) to visit grocery stores compared to before COVID-19. These models did not find a measurable advantage/disadvantage for Black populations in these places compared to White populations. COVID-19 policy should not assume disadvantage in achieving social distancing accrue equally to different racial/ethnic minorities.

## Introduction

According to data reported in the *New York Times* [1], in the United States non-Hispanic Black (henceforth Black) and Hispanic people are "three times as likely to become infected [by

Google website (https://www.google.com/covid19/mobility/). Data on violent crime can be accessed through the FBI's Uniform Crime Report site (https://www.fbi.gov/services/cjis/ucr). The social capital index can be obtained through the project website(https://aese.psu.edu/nercrd/community/social-capital-resources). Data on state and county policy dates came from the National Association of Counties website (https://www.naco.org/).

**Funding:** The authors received no specific funding for this work.

**Competing interests:** The authors have declared that no competing interests exist.

COVID-19] as their white neighbors" and "nearly twice as likely to die from the virus as white people." Physical distancing, including avoiding places with large groups of people, is the most widely recommended way to stem COVID-19's spread, and evidence suggests that barriers to achieving it may play a role in these racial and ethnic disparities [2]. Racial/ethnic inequalities undermined efforts to manage past health crises in the United States, like the HIV pandemic [3, 4]. However, the physical distancing currently being prescribed by American policy makers is unprecedented in recent times in its scale [5]. Thus it may lead to different outcomes by race/ethnicity than past events [2, 6, 7]. It is therefore vital to examine physical distancing patterns according to race and ethnicity.

This study explores how race/ethnicity relates to physical distancing in response to COVID-19 in the United States, as measured by avoidance of busy places. We take advantage of county-level social mobility data recently made available by Google [8]. This data allows us to account for the physical distancing from places that commonly attract crowds of people, including recreational sites, grocery stores, and workplaces. We focus specifically on the early stages of distancing, from February 16th to March 29th 2020. We first utilize hierarchical linear modeling (HLM) to nest counties in their metropolitan and micropolitan areas (Core-Based Statistical Areas, CBSAs) to directly determine whether racial/ethnic segregation predicts distancing behaviors [9].

HLM is limited as it assumes local effects remain within the boundaries of the units being analyzed [10]. To unpack the underlying relationship of race/ethnicity to distancing within and between counties, we employ geographically weighted regression (GWR). GWR allows us to generate empirically determined coefficients for each county weighted against the values of their neighbors, producing a more nuanced, individualized assessment of the association between race/ethnicity and COVID-19 physical distancing [10, 11]. Based on such analysis we paint a nationwide picture of the relationships between race/ethnicity and COVID-19 physical distancing.

## Distance sites

The physical distancing policies of 2020 are unprecedented in recent decades. Other recent outbreaks in America have generally had better initial screening or disproportionately affected specific populations, allowing for targeted quarantining of populations. COVID-19, meanwhile, has necessitated wide-ranging shelter-at-home policies [5]. However, these policies allow some movement based on destination. In the following paragraphs, we unpack what past research suggests about whether racial/ethnic groups are likely to avoid three kinds of sites, each with a different degree of essentiality for day-to-day life, during the pandemic: grocery stores, workplaces, and recreational sites.

Grocery stores are generally exempt from COVID-19 closure policies, though public officials have encouraged people to minimize visits and use food delivery when possible [12, 13]. Accessibility to these kinds of resources highlights racial/ethnic divisions that may impact distancing. For one, non-White communities often lack easy access to quality grocery stores with decent supplies [14], which could lead to more frequent store visits. In addition, food delivery services are less accessible in predominately Black and Hispanic areas [15, 16]. These forces could combine to make people of color more likely to visit grocery stores more frequently than non-Hispanic Whites (henceforth White).

COVID-19 physical distancing from workplaces is area where inequality by race/ethnicity is likely. While policymakers have encouraged employers to have their employees work from home, many jobs cannot be performed at home, and people of color disproportionately cannot work remotely [12, 13]. The upper income professional jobs that have flexibility to work from home are mainly staffed by Whites [17]. People of color disproportionately hold jobs in areas

such as agriculture and delivery services [18–20]. The pandemic has increased demands for workers in areas such as health care and delivery, as well [19, 21].

Whether recreational sites, including restaurants, bars, coffee shops, museums, and gyms, are likely to have disproportionate traffic is less clear. They are arguably the least "essential" places in U.S. society [22] and state and county shelter at home policies have generally closed them first and opened them last [13]. However, recreational activities at these sites are powerful determinants of social bonds and people have been known to ignore health recommendations to visit these sites because of their support for social connections [23]. It is not clear if such behavior would have racial/ethnic patterns in 2020. Anecdotal evidence reveals examples of both Black [24, 25] and White communities [26] ignoring COVID-19 distancing policies to see friends and family.

Racial/ethnic enclaves, including recreational sites, may be a particular draw to racial/ethnic minorities. Past research shows these areas are of unique importance for some racial/ethnic groups to build connections and avoid discrimination from outsiders [27]. People of color will inconvenience themselves to go to such places [28]. This raises the question as to whether they will also take the risk of infection.

## The inclination to distance

Resistance to distancing may be greater in non-White communities than in White communities. Non-White communities have demonstrated a deep distrust of advice from the government and health care professionals [29]. This distrust is a consequence of the long history of discriminatory behavior towards people of color from these entities. The government and public health system ignored the HIV epidemic because many of the early patients were Black [4, 30] and mismanaged disasters like Hurricane Katrina [31]. History such as the Tuskegee Syphilis experiments on unknowing Black men have further eroded trust [30]. Hispanic populations have their own reasons to distrust the government and health care system. Hispanics face discrimination due to their real or presumed undocumented status [32, 33], and there is a history of syphilis experimentation on this community as well [34]. Both Black and Hispanic populations often endure condescending and generally unhelpful behavior from medical practitioners, and research shows this exacerbates their distrust of health care advice [35]. Likewise, there is early evidence of racial/ethnic bias in efforts to enforce COVID-19 distancing, which further erodes trust in government. For example, police in New York City have disproportionately targeted Black and Hispanic populations who have failed to distance [36, 37], leading some in the media to decree physical distancing policy the "new stop and frisk" [38]. These factors may prompt unwillingness to distance among people of color.

At the same time, White populations may resistance distancing as well. Trust in the U.S. government is at a historic low among the population as a whole [39]. The anti-vaccination movement, which disproportionately attracts middle and upper income Whites, also suggests White people have low trust in health care and public health professionals [40]. In addition, the growing splintering of news media and community, as well as inconsistent recommendations from the U.S. government, has propagated misinformation since the pandemic began has disproportionately influenced dismissals of the dangers of COVID-19 [41]. Thus, White populations may be just as unlikely to distance as Black and Hispanic populations in response to COVID-19.

## Segregation and community

Another factor in understanding race/ethnicity that can relate both to the ability and willingness of people to distance in response to COVID-19 is residential segregation based on race/

ethnicity. Segregation is the result of the resistance of White populations to live next to non-Whites and formal and informal efforts from different levels of government, the housing market, and banking infrastructure to support this resistance [42, 43]. Segregation has many dimensions; isolation, the chance two randomly selected residents in a neighborhood will be of the same race/ethnicity, is a useful measure of it as it reveals social environment [44].

The existing literature shows that the homogenous social environments resulting from segregation present complications that are likely to influence how race/ethnicity predicts COVID-19 distancing. On the one hand, isolated communities are sites of considerable social and economic inequalities as a consequence of systemic discrimination, including a lack of resources, which directly impacts preparations for health crises [45, 46]. These disparities can exacerbate people of color's existing distrust of the health care system through providing a demonstration as for why these institutions cannot be trusted [4, 30]. On the other hand, isolated communities may have the counterintuitive effect of *increasing* the chance of COVID-19 distancing among people of color. Some studies show that segregation can increase the likelihood of following health recommendations. Black residents in predominately Black neighborhoods have greater trust in the health care system than Blacks in more mixed neighborhoods [29]. Areas with high shares of Black residents often have cultural hubs for these groups, like churches and barbershops, that shield them from discrimination, encouraging the spread, and adherence, to medical advice [47, 48]. Moreover, these sites can be used to help people of color select healthcare providers who are less discriminatory, which can further improve trust in the healthcare system as a whole [49]. There is a risk this support system can backfire, for example non-White communities have been known to spread incorrect facts about how HIV is transmitted, which had a documented effect on it's spread and mobility [3]. Also, not all mostly Black neighborhoods have these cultural hubs, meaning the positive association of percent Black to healthy behaviors may vary by location [50].

The strength of community ties drive whether segregation reduces health care system distrust in non-White communities [29]. It is through close social ties that trust can emerge [51]. The strength of this community support can vary from place to place. The economic disadvantage found in some segregated racial/ethnic minority communities can erode the cumulative effectiveness of local social ties [52]. How severe this erosion, and how it matters for healthy preparedness, can vary. There is some evidence that low-income Hispanic communities have superior health outcomes compared to other, similarly low-income racial/ethnic communities [53]. For example, during the 1995 Chicago heat wave, Black and Hispanic neighborhoods of a similar socioeconomic character diverged in their ability to lessen the effects of the heat [50]. The Hispanic Chicago neighborhood had a strong community and civic connections, fortified by familial ties, which helped them to navigate the crisis. The Black neighborhood, meanwhile, was not only poor but had endured a long-term community decline that diminished their social infrastructure. Whether communities will show similar patterns during the COVID-19 pandemic is unclear.

However, past research on differential impacts according to the prevalence of particular racial/ethnic groups is too patchy to make any assumptions about what analysis concerning the entire United States will reveal. For example, the 1995 heatwave study only focused on poor Black and Hispanic neighborhoods in Chicago [50], a city that has high residential segregation for Blacks and Hispanics. Residential segregation has different magnitudes for racial/ethnic groups across the United States. Hispanic segregation, for example, is more prevalent in the Southwest while Black segregation is more prevalent in the Northeast, Midwest, and South [54]. Also, the type of distancing may vary in importance depending on the site in which it is taking place.

## Social capital

Another consideration with social distancing is the presence of social capital [55], the sum benefits of social connections for a community [56]. Racial/ethnic concentrations are a unique source of social capital as they facilitate a sense of solidarity through shared lived experiences [57]. Past research suggests social capital in ethnically concentrated locations can have either a positive or negative association with distancing practices. On the one hand, robust social capital allows the spread of information and resources to facilitate healthy behaviors in non-White communities [58–60], such as distancing. However, the effectiveness of social capital is only as strong as the residents from whom it is composed [61]. As such, on the other hand, the information and resources being shared may instead discourage distancing practices [23]. For example, Black communities in particular have a longstanding distrust of healthcare that can be reinforced through strong community bonds [62].

## Spatial heterogeneity

One final consideration in evaluating how racial/ethnic populations relate to physical distancing is the local variation of context. Research exploring the spatial character of the relationship of race and ethnicity to various outcomes has shown this relationship is not consistent across all places even if the share of these populations and other relevant factors like socio-economic status are controlled for [10, 63, 64]. Also, not all mostly Black areas will have the same attachment to their locations due to highly specific localized factors. For example, some Black areas may be more effective at sharing health information, such as the importance of avoiding busy places, due to the personal opinions of key social actors within these places [65]. Following on this, the community dynamics of Black areas in highly segregated regions may vary greatly from place to place [9]. These small factors, which cannot be directly modelled for in non-spatial models, can have an important local effect on race and health behaviors which can be glossed over [63, 64].

## Hypotheses

Past research is instructive in charting possible associations of racial/ethnic populations to COVID-19 physical distancing. Racial/ethnic disparities may inhibit the ability and willingness these groups to distance in response to this pandemic. Alternatively, while residential segregation in many ways exacerbates racial/ethnic disparities, may create the infrastructure to encourage distancing in response to COVID-19. Past work leaves open several questions. First, do both Black and Hispanic populations have the same level of difficulty in distancing in response to COVID-19? Second, how consistently does race/ethnicity relate to these different distancing sites across the United States? To guide this study, we draw upon the following hypotheses. Most of the past evidence points to the challenges non-White populations face in following health advice [4, 15, 16, 18–21, 29, 66]. We hypothesize:

H1. *Regardless of the type of COVID-19 physical distancing, racial/ethnic minority populations will be less likely to distance than Whites.*

The literature offers reason to suspect both Black and Hispanic populations have greater trouble physical distancing in response to COVID-19 than White populations. Past research suggests Black populations will have greater trouble than Hispanic ones. We hypothesize:

H2. *Regardless of the type of COVID-19 physical distancing, Black populations will be less likely to physically distance than Hispanic populations.*

We also believe residential segregation will relate to COVID-19 non-White physical distancing [29, 45, 46, 49]:

*H3. Higher racial/ethnic segregation relates to more COVID-19 physical distancing for racial/ethnic minority populations.*

Lastly, it is likely the impact of race/ethnicity will not be consistent across the U.S.. We hypothesize:

*H4. The association of race/ethnicity to COVID-19 physical distancing will not be consistent across the United States.*

## Data and measures

We draw upon county-level mobility data from Google [8] to measure COVID-19 distancing from *grocery*: grocery stores, food warehouses, farmers markets, specialty food shops, drug stores, and pharmacies; *recreation*: restaurants, cafes, shopping centers, theme parks, museums, libraries, and movie theaters; and *workplaces*. These three locations were chosen in part because the other locations available from Google had considerable missing data. Google has established through proprietary methods the difference in regular daily movements of users of their Google Maps software to and from these locations. They established a baseline of pre-COVID-19 movements from January 3rd to February 6th 2020. From there, they determine how the regular movements to these places differ from the baseline from February 16th to March 29th 2020. The resulting value is a percentage score reporting the difference between the COVID-19 period and the baseline. The study period captures the early stages of the American response to COVID-19, pointing to the initial disparities in the capacity for racial/ethnic groups to avoid the above locations. With its extensive geographic database, Google establishes whether a location is a recreational site, grocery, or workplace [67]. This categorization is specific to the user of Google Maps, which allows users to save their "workplace" for easy accessing. A grocery store would be identified as a "workplace" if a user designates it their workplace and as a "grocery" for all others. The resulting data is aggregated account for these various nuances of movement [68]. Google determined counties with low counts and omitted them [8]. Despite these limitations, a recent study [55] reports that the Google data yields results comparable with other data sources, which offers a certain level of external validity to the Google data.

There are some important considerations with this data. Many important aspects of this data, such as how Google categorizes locations, is based on proprietary information. This makes it difficult to validate their designation of sites. The exact number of Google Maps users is also proprietary. While existing estimates put the number of regular Google Maps users into the billions [69], their distribution geographically and in terms of race and ethnicity is unclear. Of greatest pertinence to this study, Google does not disaggregate its data by racial/ethnic groups, meaning this study must use racial/ethnic breakdown of residents as a proxy to determine the behavior of Blacks, Hispanics, and Whites.

We measure the racial/ethnic distribution of counties two ways. First, to measure the racial/ethnic composition of a county, we use the 2018 American Community Survey to identify the percentage non-Hispanic White (reference), non-Hispanic Black, Hispanic, and Asians. Asians are non-Whites but their infection and death rate from COVID-19 in the United States is far lower than Blacks and Hispanics [1]. This measure allows us to roughly proxy how racial/ethnic groups are distancing by county–for example Black population size negatively relating to distancing would suggest this is connected to the Black population. However, composition alone does not directly measure the segregated social environment of people of color. This leads to our second measure of racial/ethnic distribution, the Spatial Isolation Index, the probability two randomly selected individuals in a CBSA being of the same race/ethnicity. We

focus on isolation as opposed to exposure, the chance a racial/ethnic minority encounters a White person, in light of isolation's emphasis on racially/ethnically distanced social environments [9]. The Spatial Isolation Index is a more robust measure of segregation than racial/ethnic composition alone. Spatial Isolation can be decomposed with the change in the boundaries of subareas and the decomposed values are additive. In addition, it can be applied to both aggregated population counts (zone-based) or continuous population density (surface-based). This allows it to measure segregation within and between counties in a CBSA. We emphasize two group measures of Spatial Isolation as they are more suited to show how isolation from Whites, specifically, relates to individuals' ability to distance in response to COVID-19. We chose CBSAs as our level-two measure as their components, metropolitan and micropolitan areas, correspond to counties and are a common level at which to measure segregation [54, 70]. Segregation is only used in the HLM model as we cannot easily control for CBSA fixed effects in a GWR model.

To determine the willingness and ability to distance in response COVID-19, we draw upon Andersen's behavioral model [10, 71], which details the "enabling factors," which tie into the ability to distance, and "predispositions," the willingness to distance. County-level predisposing factors using the ACS include education, measured as *percent college educated*, and age, which we measure as the percent of adults over the *age of 65*. Race/ethnicity, already discussed, is also considered a "predisposing" factor. Enabling factors include socioeconomic status, including the *percent living in poverty*, *percent with health insurance*, and *percent unemployed*. As part of enabling factors, we also measure percent of workers with a *long commute*, commuting more than 45 minutes, and *population density*, based on their association with accessibility of health services [72]. We assess residential stability with percent homeowners and percent who have lived in their current homes for five years or less [73]. We had also sought to also measure percent foreign born at the county-level but were unable to include this measure due to collinearity.

In addition to the ACS, we also draw upon the *Social Capital Index*, developed by Rupasingha Goetz and Freshwater [74], as another county-level enabling factor. This index is generated from a principal component analysis (PCA) of civic organizations and associations (religious, civic, businesses, political, professional, labor, sports, and miscellaneous recreation) and nonprofit organizations, as well as voter turnout and the Census response rate. The resulting PCA score is standardized, with a mean of zero and standard deviation of approximately 1. Another enabling factor is speed of government policy in response to COVID-19. Using the National Association of Counties County Explorer [13], we establish the number of days since states and counties enacted a shelter policy, starting March 19[th] 2020 in the case of states and March 16[th] 2020 in the case of counties. States and counties that did not enact a policy by March 29[th] were given the minimum value for each, no days. Lastly, given the potential of crime to impact the willingness of people to go out, we measure the share of violent crimes per 100,000, based on the Uniform Crime Report as another enabling factor.

We avoided using COVID-19 mortality data and hospitalization rates due to considerable inconsistencies in this data [75]. The final data sets, broken down by outcome, included 2,258 counties (526 CBSAs) for grocery visits, 2,345 counties (535 CBSAs) for recreation visits, and 2,474 counties (544 CBSAs) for workplace visits.

## Methods

This study draws on two analytical techniques to determine the association of local context to the racial/ethnic variation in physical distancing in response to COVID-19. We employ HLM using the R package *lme4* to determine how specific counties relate to the CBSA context.

Through this method, we can nest counties within their respective CBSAs to determine whether they explain some of the variation in the distancing within these counties. However, HLM is limited in that it treats local effects as stationary and mutually independent across places [10]. To better establish how the presence and underlying characteristics of COVID-19 physical distancing varies across space, we supplement these models with GWR using the *spgwr* R package.

GWR is a tool that allows us not only to better understand the relationships between race, space, and COVID-19 physical distancing suggested in the HLM models, but also to unpack these relationships in ways HLM models cannot due to their stationary character. GWR determines the spatial variation of the parameters for each physical distancing outcome across the contiguous 48 states [11]. While there has been some debate about the strength of GWR estimates, several recent studies have demonstrated it is a robust way to generate local estimations [76–79]. To generate local estimates, GWR uses a "moving window" weighting strategy wherein the parameters for each county are weighted against the parameters for nearby counties. The sensitivity of this weighting is determined by a bandwidth, which is established based on modeling strategy. The estimation process is applied to all counties and the resulting GWR coefficients can be interpreted in the same manner as HLM coefficients [11, 80]. For example, the percent Black coefficient in the workplace model indicates the relationship one percent change in percentage of Black people has with the level of difference in regular movement to workplaces during the COVID-19 outbreak compared to the baseline. As the GWR model generates results for each county in our data, we use two methods to describe our results: First, we reported summary statistics (minimum, three quartiles, and maximum) of the local coefficient estimates. Second, we mapped out select coefficients for each distancing site analyzed.

## Results

We begin by discussing the descriptive values of data discussed in Table 1, broken down into three data sets by outcome. The average county practiced physical distancing, according to our three distancing measures. The greatest distancing is from recreational sites, with a difference of -38.5 percent from the baseline. The smallest degree of distancing is grocery stores, with a value of -14.316. However, the standard deviation for grocery distancing is the largest relative to its mean compared to the other distancing values, greater than the mean at 15.144, meaning the nature of grocery distancing varies considerably across counties. To better visualize this variation, we map the distancing values in Fig 1. We can see that while each distancing by site varies in its success, some rough patterns emerge. Distancing is generally more successful in coastal regions, where large United States cities can be found. Distancing, regardless of site, is generally less successful in the South. Work distancing differs from these trends; unlike recreation and grocery distancing, it is comparatively less likely to be successful in the Midwest.

For the other predictors, we report on the values from the Work distancing data set to streamline the discussion. The variation of predictors between these data sets is comparatively small. A large share of counties are White, on average 76.766 percent. The non-White populations are approximately 10 percent Black, 10 percent Hispanic, and about 1.5 percent Asian. The socioeconomic indicators are divided in these counties. While 22.1 percent are college educated, 12.5 percent live in poverty. Most have some form of health insurance, 90.3 percent. Meanwhile, most (84.57) have a commute of 45 minutes or longer. As for our measure of spatial isolation, the chance of a Black person encountering another Black person is 14.2 percent (0.142). However, the maximum value for this score is 0.713, or 71.3 percent. While the average Hispanic value is similar to the average Black value, the maximum Hispanic value is only 46.1 percent.

**Table 1. Descriptive values by social distance site.**

| Statistic | Workplace | | | | Recreation | | | | Grocery | | | |
|---|---|---|---|---|---|---|---|---|---|---|---|---|
| | Mean | St. Dev. | Min | Max | Mean | St. Dev. | Min | Max | Mean | St. Dev. | Min | Max |
| *County-Level* | | | | | | | | | | | | |
| Distancing Type | -30.890 | 8.862 | -76.000 | 3.000 | -38.450 | 16.888 | -100.000 | 133.000 | -14.316 | 15.144 | -73.000 | 128.000 |
| Percent White | 76.766 | 19.077 | 0.728 | 99.589 | 76.525 | 18.992 | 0.728 | 99.396 | 75.756 | 19.235 | 0.728 | 99.396 |
| Percent Black | 9.025 | 13.741 | 0.000 | 87.412 | 9.008 | 13.476 | 0.000 | 87.412 | 9.352 | 13.638 | 0.000 | 81.783 |
| Percent Asian | 1.425 | 2.518 | 0.000 | 35.650 | 1.483 | 2.572 | 0.000 | 35.650 | 1.517 | 2.610 | 0.000 | 35.650 |
| Percent Hispanic | 9.350 | 13.452 | 0.000 | 99.069 | 9.615 | 13.640 | 0.000 | 99.069 | 9.973 | 14.152 | 0.000 | 99.069 |
| Percent Female | 50.060 | 2.025 | 33.389 | 55.069 | 50.078 | 2.015 | 33.389 | 55.069 | 50.092 | 2.040 | 33.389 | 55.069 |
| Percent College Educated | 22.097 | 9.665 | 5.380 | 74.561 | 22.418 | 9.742 | 6.882 | 74.561 | 22.424 | 9.842 | 5.380 | 74.561 |
| Percent 64 or Older | 17.905 | 4.215 | 3.799 | 55.596 | 17.799 | 4.243 | 3.799 | 55.596 | 17.663 | 4.219 | 3.799 | 55.596 |
| Percent Poverty | 12.430 | 4.728 | 1.671 | 35.609 | 12.356 | 4.646 | 1.671 | 33.851 | 12.444 | 4.669 | 3.023 | 35.609 |
| Percent Unemployed | 3.306 | 1.149 | 0.400 | 14.990 | 3.321 | 1.105 | 0.499 | 8.774 | 3.345 | 1.106 | 0.000 | 8.774 |
| Percent Insured | 90.254 | 4.705 | 60.455 | 98.198 | 90.266 | 4.681 | 60.455 | 98.198 | 90.148 | 4.726 | 60.455 | 98.198 |
| Percent Long Commute | 84.570 | 7.821 | 48.249 | 98.122 | 84.739 | 7.702 | 48.249 | 98.122 | 84.641 | 7.755 | 48.249 | 98.122 |
| Percent Moved | 4.226 | 1.474 | 0.602 | 14.604 | 4.284 | 1.470 | 0.809 | 14.604 | 4.303 | 1.475 | 0.602 | 14.604 |
| Violent Crime Per 100,000 | 265.371 | 187.662 | 0.000 | 1378.436 | 271.785 | 186.797 | 0.000 | 1378.436 | 275.320 | 187.795 | 0.000 | 1378.436 |
| Social Capital Index | -0.196 | 1.017 | -3.183 | 21.809 | -0.210 | 0.997 | -3.183 | 21.809 | -0.265 | 0.960 | -3.183 | 21.809 |
| County Policy | 8.000 | 3.294 | 0.000 | 11.000 | 8.000 | 3.327 | 0.000 | 11.000 | 8.000 | 3.334 | 0.000 | 11.000 |
| State Policy | 13.000 | 1.687 | 0.000 | 14.000 | 13.000 | 1.729 | 0.000 | 14.000 | 13.000 | 1.750 | 0.000 | 14.000 |
| Population Density | 0.000 | 0.001 | 0.000 | 0.028 | 0.000 | 0.001 | 0.000 | 0.028 | 0.000 | 0.001 | 0.000 | 0.028 |
| *CBSA-Level* | | | | | | | | | | | | |
| White/Black Isolation | 0.142 | 0.141 | 0.000 | 0.713 | 0.144 | 0.141 | 0.000 | 0.713 | 0.147 | 0.141 | 0.000 | 0.713 |
| White/Hispanic Isolation | 0.136 | 0.149 | 0.000 | 0.993 | 0.139 | 0.152 | 0.000 | 0.993 | 0.143 | 0.156 | 0.000 | 0.993 |
| White/Asian Isolation | 0.041 | 0.051 | 0.000 | 0.461 | 0.042 | 0.052 | 0.000 | 0.461 | 0.042 | 0.053 | 0.000 | 0.461 |
| | 2,474 | | | | 2,345 | | | | 2,258 | | | |

To visualize segregation, in Fig 2 we map out the values by CBSA and compare them to the county-level values of percent race/ethnicity. We find clear spatial patterns to segregation–Blacks are most likely to encounter other Blacks across the South. High isolation values roughly correspond to the counties with the highest share of Blacks, which are disproportionately in the South. Meanwhile, the highest Hispanic isolation can be found in the Southwest. Asian isolation is strongest in the West and parts of the Northeast, though the magnitude of the Asian segregation score is lower than that of Black and Hispanic populations.

## HLM results

To determine whether services are related to connectivity, we turn to our HLM models. To ensure the appropriateness of the HLM approach, we first conducted a null model where no independent variable is included and determined whether the distancing measures differ by CBSAs. We find all of the distancing measures are significant, warranting an HLM approach. To determine the overall influence of CSBAs on these measures, we conduct intraclass correlation coefficient (ICC) tests. We find the influence CBSAs is high for each type of distancing–CBSAs explain 35.5 percent of the variation in work distancing, 37.6 percent of recreation distancing, and 44.7 percent of grocery distancing.

We start by discussing the county level HLM results for each outcome, reported in Table 2. When interpreting the coefficients, we will refer to negative values in terms of "more" or "greater" distancing and positive values as "less" or "weaker" distancing. Only percent Asian

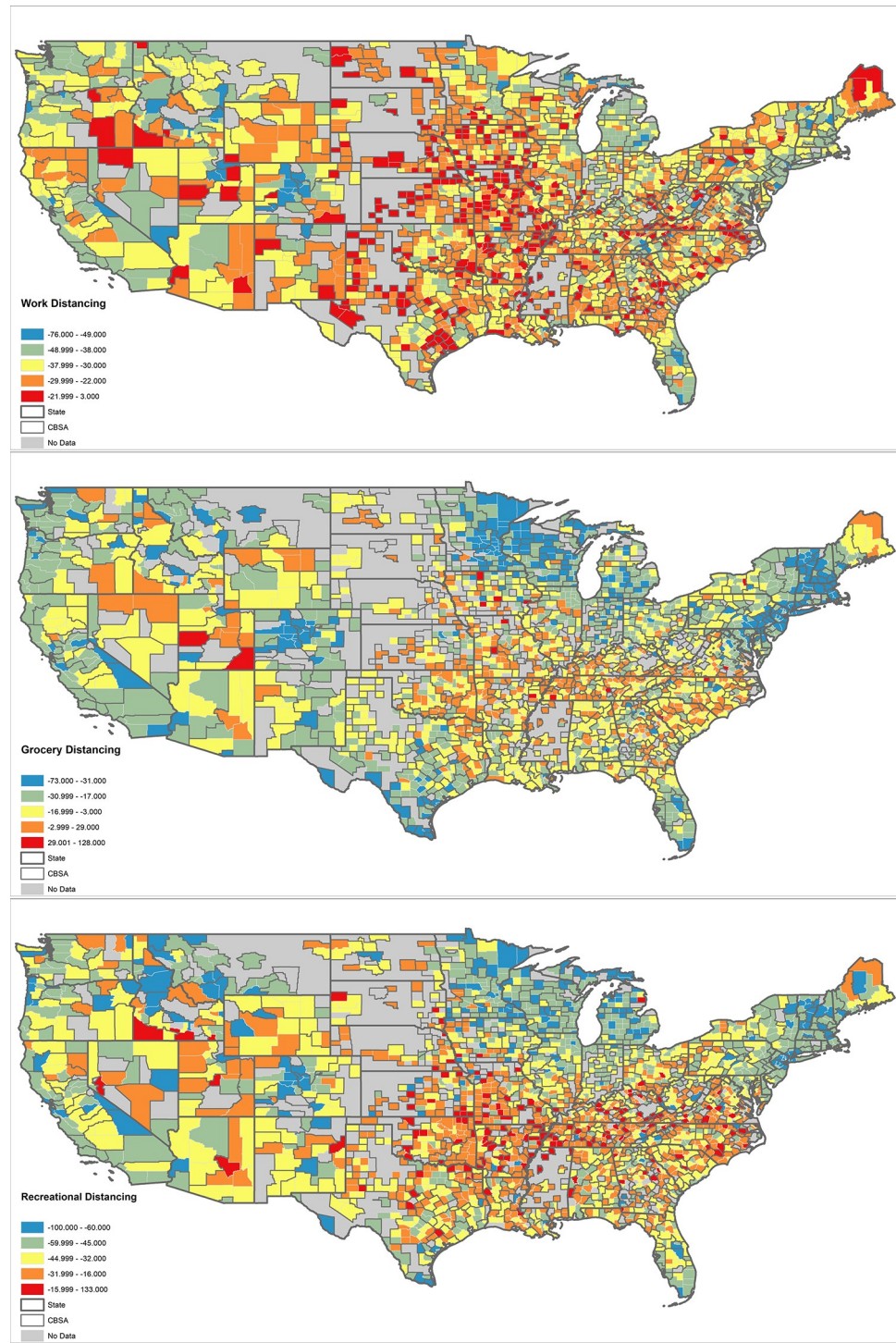

**Fig 1. Descriptive values of social distancing.** Source TIGER/line.

has a significant relationship with less distancing, specifically to workplace distancing. Meanwhile, the share of Hispanic populations is related to more distancing for both recreation and grocery distancing. For example, with each point increase in the percent Hispanic in the

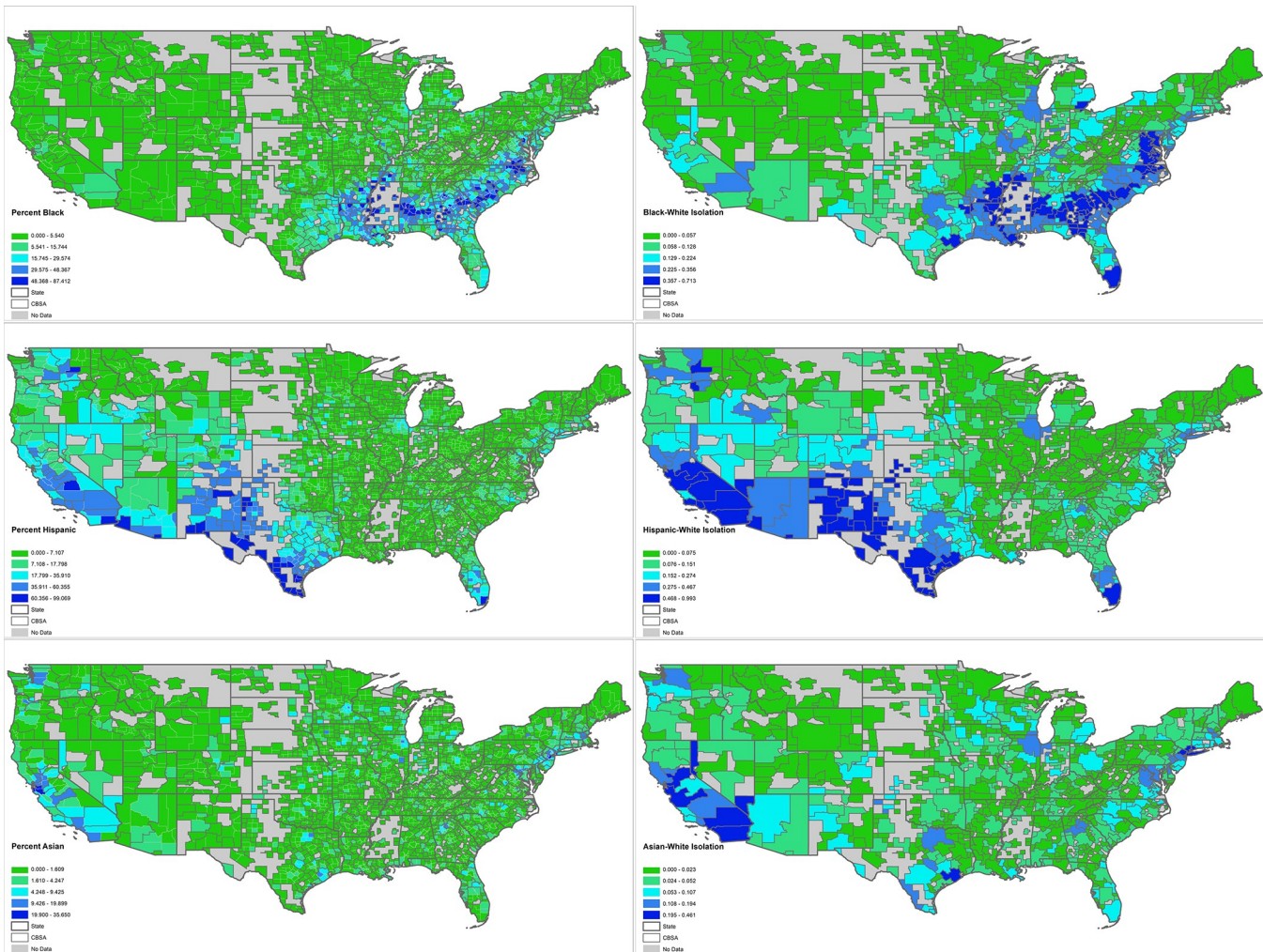

**Fig 2. Descriptive values of racial/ethnic groups.** Source TIGER/line. Note: Cut points determined with natural breaks.

White/Black model, the decline in regular visits to recreational sites since the start of COVID-19 increases by 0.055 points. We find that isolation's most significant coefficient is White/Black segregation in the recreation model. Each point increase of the isolation index is related to an 11.476 point reduction in county-level COVID-19 distancing. To explore this association further, we conducted supplemental cross-level analysis (available upon request) interacting percent Black or percent White with the isolation index. While neither interaction term was significant, the significance of the isolation index was lost with the inclusion of the percent White X isolation index term–suggesting the percent White is driving this effect. To better understand the relationship of local context to distancing, we report on our GWR results. When evaluating the other predictors, the most consistent coefficients are county and state shelter at home policies.

## GWR results

Given the GWR models separate results for all counties for each outcome, a five-value summary is the most efficient means of reporting the range of coefficients, reported in Table 3. An Akaike Information Criterion (AIC) was used to determine whether the GWR model fit the

**Table 2. HLM social distancing by site.**

| | Workplace | | | Recreation | | | Grocery | | |
|---|---|---|---|---|---|---|---|---|---|
| | Black/ White | Hispanic/ White | Asian/ White | Black/ White | Hispanic/ White | Asian/ White | Black/ White | Hispanic/ White | Asian/ White |
| *County-Level* | | | | | | | | | |
| Percent Black | -0.0001 | 0.008 | 0.008 | -0.029 | 0.044 | 0.046 | -0.060 | -0.017 | -0.016 |
| | (0.020) | (0.016) | (0.016) | (0.044) | (0.034) | (0.034) | (0.039) | (0.031) | (0.031) |
| Percent Asian | 0.168** | 0.163** | 0.153* | 0.191 | 0.181 | 0.204 | 0.220 | 0.220 | 0.254* |
| | (0.077) | (0.077) | (0.081) | (0.157) | (0.157) | (0.165) | (0.139) | (0.139) | (0.145) |
| Percent Hispanic | 0.003 | -0.031 | 0.001 | -0.055* | -0.092* | -0.053 | -0.169*** | -0.127*** | -0.162*** |
| | (0.015) | (0.024) | (0.015) | (0.031) | (0.049) | (0.032) | (0.029) | (0.043) | (0.030) |
| Percent Female | 0.211*** | 0.220*** | 0.214*** | 0.829*** | 0.869*** | 0.865*** | 0.103 | 0.113 | 0.124 |
| | (0.073) | (0.072) | (0.073) | (0.154) | (0.154) | (0.154) | (0.137) | (0.137) | (0.137) |
| Percent College Educated | -0.544*** | -0.549*** | -0.544*** | -0.742*** | -0.745*** | -0.740*** | -0.565*** | -0.556*** | -0.562*** |
| | (0.023) | (0.023) | (0.023) | (0.049) | (0.049) | (0.049) | (0.044) | (0.045) | (0.044) |
| Percent 64 or Older | -0.208*** | -0.224*** | -0.210*** | -0.240*** | -0.262*** | -0.244****** | 0.044 | 0.062 | 0.044 |
| | (0.043) | (0.043) | (0.043) | (0.089) | (0.090) | (0.089) | (0.080) | (0.082) | (0.080) |
| Percent Poverty | 0.078 | 0.079* | 0.078 | 0.258** | 0.243** | 0.236** | 0.064 | 0.049 | 0.045 |
| | (0.048) | (0.048) | (0.048) | (0.102) | (0.102) | (0.102) | (0.092) | (0.092) | (0.092) |
| Percent Unemployed | -0.783*** | -0.788*** | -0.787*** | -0.075 | -0.081 | -0.069 | 0.075 | 0.079 | 0.092 |
| | (0.148) | (0.148) | (0.148) | (0.324) | (0.325) | (0.325) | (0.288) | (0.288) | (0.289) |
| Percent Insured | 0.130*** | 0.137*** | 0.128*** | 0.299*** | 0.290*** | 0.281*** | -0.043 | -0.067 | -0.054 |
| | (0.043) | (0.043) | (0.043) | (0.091) | (0.091) | (0.091) | (0.083) | (0.084) | (0.083) |
| Percent Long Commute | 0.135*** | 0.139*** | 0.136*** | -0.029 | -0.041 | -0.051 | -0.040 | -0.059 | -0.058 |
| | (0.021) | (0.021) | (0.021) | (0.045) | (0.045) | (0.046) | (0.040) | (0.041) | (0.041) |
| Percent Moved | -0.058 | -0.069 | -0.059 | 0.645*** | 0.621*** | 0.632*** | -0.189 | -0.183 | -0.199 |
| | (0.113) | (0.113) | (0.113) | (0.238) | (0.238) | (0.238) | (0.214) | (0.214) | (0.214) |
| Violent Crime Per 100,000 | -0.00004 | -0.0001 | -0.0001 | -0.006*** | -0.006*** | -0.006*** | 0.002 | 0.002 | 0.002 |
| | (0.001) | (0.001) | (0.001) | (0.002) | (0.002) | (0.002) | (0.002) | (0.002) | (0.002) |
| Social Capital Index | 0.681*** | 0.676*** | 0.670*** | -1.313*** | -1.400*** | -1.403*** | 0.004 | -0.052 | -0.043 |
| | (0.169) | (0.168) | (0.169) | (0.356) | (0.355) | (0.355) | (0.327) | (0.326) | (0.326) |
| County Policy | 0.492*** | 0.507*** | 0.510*** | 0.560* | 0.594* | 0.564* | -0.261 | -0.289 | -0.260 |
| | (0.160) | (0.160) | (0.162) | (0.336) | (0.337) | (0.341) | (0.278) | (0.278) | (0.278) |
| State Policy | 1.702*** | 1.720*** | 1.728*** | 3.820*** | 3.892*** | 3.854*** | 2.880*** | 2.903*** | 2.850*** |
| | (0.196) | (0.195) | (0.198) | (0.417) | (0.416) | (0.423) | (0.398) | (0.398) | (0.404) |
| Population Density | -0.366** | -0.352** | -0.369** | -0.780** | -0.772** | -0.785** | 0.451 | 0.450 | 0.421 |
| | (0.150) | (0.150) | (0.150) | (0.312) | (0.313) | (0.313) | (0.306) | (0.306) | (0.310) |
| *CBSA-Level* | | | | | | | | | |
| Isolation | 1.342 | 4.262* | 3.009 | 11.476*** | 4.578 | -3.770 | 7.493* | -5.884 | -10.700 |
| | (2.072) | (2.272) | (5.243) | (4.383) | (4.698) | (11.026) | (4.149) | (4.401) | (10.782) |
| Constant | -47.341*** | -48.367*** | -47.179*** | -89.100*** | -87.753*** | -86.091*** | -0.019 | 3.819 | 2.474 |
| | (5.592) | (5.612) | (5.577) | (11.872) | (11.935) | (11.856) | (10.816) | (10.859) | (10.784) |
| Observations | 2,474 | 2,474 | 2,474 | 2,345 | 2,345 | 2,345 | 2,258 | 2,258 | 2,258 |

(*Continued*)

**Table 2.** (Continued)

| | Workplace | | | Recreation | | | Grocery | | |
|---|---|---|---|---|---|---|---|---|---|
| | **Black/ White** | **Hispanic/ White** | **Asian/ White** | **Black/ White** | **Hispanic/ White** | **Asian/ White** | **Black/ White** | **Hispanic/ White** | **Asian/ White** |
| AIC. | 16,411.700 | 16,408.420 | 16,409.940 | 18,870.890 | 18,876.640 | 18,875.770 | 17,643.170 | 17,644.530 | 17,643.540 |

Notes

*p<0.1

**p<0.05

***p<0.01

#: Standardized.

data better than the HLM model [11]. As a rule of thumb, when the difference in AICs between two models is larger than 4, the model with the smaller AIC is strongly preferred [81]. We find that all AIC values for the GWR models are much smaller than their HLM equivalents. For example, the HLM Grocery AIC is 18157.43 while the GWR AIC is 17382.26, indicating GWR better fits the data set.

**Table 3. GWR social distancing by type.**

| | Workplace | | | | | Recreation | | | | | Grocery | | | | |
|---|---|---|---|---|---|---|---|---|---|---|---|---|---|---|---|
| | **Min.** | **1st** | **Median** | **3rd** | **Max.** | **Min.** | **1st** | **Median** | **3rd** | **Max.** | **Min.** | **1st** | **Median** | **3rd** | **Max.** |
| Intercept | -153.210 | -68.061 | -37.412 | -14.128 | 50.463 | -244.770 | -127.710 | -96.551 | -64.920 | 6.439 | -200.580 | -33.369 | -0.350 | 28.958 | 145.620 |
| Percent Black | -0.715 | -0.086 | -0.012 | 0.044 | 1.030 | -1.061 | -0.095 | 0.011 | 0.187 | 1.015 | -0.992 | -0.176 | -0.052 | 0.075 | 0.379 |
| Percent Asian | -3.259 | -0.388 | -0.029 | 0.231 | 2.459 | -1.904 | -0.775 | -0.247 | 0.025 | 0.485 | -5.298 | -1.260 | -0.440 | 0.158 | 1.349 |
| Percent Hispanic | -0.493 | -0.093 | -0.006 | 0.092 | 0.539 | -0.584 | -0.162 | -0.056 | 0.138 | 0.699 | -0.653 | -0.292 | -0.166 | 0.095 | 0.848 |
| Percent Female | -0.965 | -0.140 | 0.120 | 0.526 | 1.855 | -0.418 | 0.375 | 0.751 | 1.233 | 2.669 | -1.472 | -0.514 | -0.109 | 0.358 | 2.045 |
| Percent College Educated | -0.983 | -0.512 | -0.398 | -0.295 | 0.022 | -1.162 | -0.759 | -0.617 | -0.484 | 0.035 | -1.019 | -0.538 | -0.349 | -0.175 | 0.408 |
| Percent Age 64 or Older | -0.927 | -0.327 | -0.182 | -0.018 | 0.373 | -0.952 | -0.434 | -0.203 | 0.011 | 0.599 | -0.642 | -0.054 | 0.305 | 0.633 | 1.970 |
| Percent Poverty | -0.701 | 0.045 | 0.236 | 0.388 | 1.240 | -0.573 | 0.085 | 0.297 | 0.781 | 2.029 | -1.276 | -0.098 | 0.143 | 0.398 | 2.850 |
| Percent Unemployed | -2.776 | -1.135 | -0.551 | -0.092 | 1.453 | -2.348 | -1.120 | -0.264 | 0.782 | 2.812 | -2.622 | -0.208 | 0.417 | 0.870 | 3.076 |
| Percent Insured | -0.563 | -0.143 | 0.008 | 0.221 | 1.228 | -0.539 | 0.105 | 0.403 | 0.629 | 1.379 | -1.223 | -0.189 | 0.005 | 0.196 | 1.112 |
| Percent Long Commute | -0.108 | 0.048 | 0.087 | 0.158 | 0.342 | -0.324 | -0.134 | -0.035 | 0.031 | 0.224 | -0.445 | -0.159 | -0.037 | 0.061 | 0.562 |
| Percent Moved | -1.286 | -0.461 | -0.156 | 0.156 | 1.254 | -1.201 | -0.022 | 0.501 | 1.206 | 2.937 | -2.512 | -1.032 | -0.533 | -0.017 | 1.626 |
| Violent Crime Per 100,000 | -0.009 | -0.002 | 0.001 | 0.003 | 0.015 | -0.030 | -0.010 | -0.004 | 0.001 | 0.010 | -0.014 | -0.004 | 0.001 | 0.007 | 0.023 |
| Social Capital Index | -2.434 | -0.013 | 0.670 | 1.618 | 4.472 | -4.431 | -2.288 | -1.214 | -0.398 | 3.400 | -5.498 | -1.735 | -0.508 | 0.368 | 2.752 |
| County Shelter at Home Policy Date | -1.786 | -0.065 | 0.323 | 0.663 | 2.485 | -3.168 | -0.066 | 0.600 | 1.176 | 4.862 | -1.922 | 0.216 | 1.033 | 1.750 | 4.080 |
| State Shelter at Home Policy Date | -1.600 | 0.612 | 1.411 | 2.155 | 7.097 | -0.031 | 2.407 | 3.419 | 4.587 | 8.824 | -1.172 | 1.556 | 2.833 | 4.634 | 13.316 |
| Population Density | -16.212 | -3.373 | -0.803 | 0.054 | 5.557 | -12.230 | -2.954 | -0.691 | 0.084 | 6.065 | -13.382 | -0.407 | 1.197 | 4.846 | 10.988 |
| AIC | 15621.230 | | | | | 18323.410 | | | | | 17082.730 | | | | |

Note: CBSA-level measures not included in GWR model.

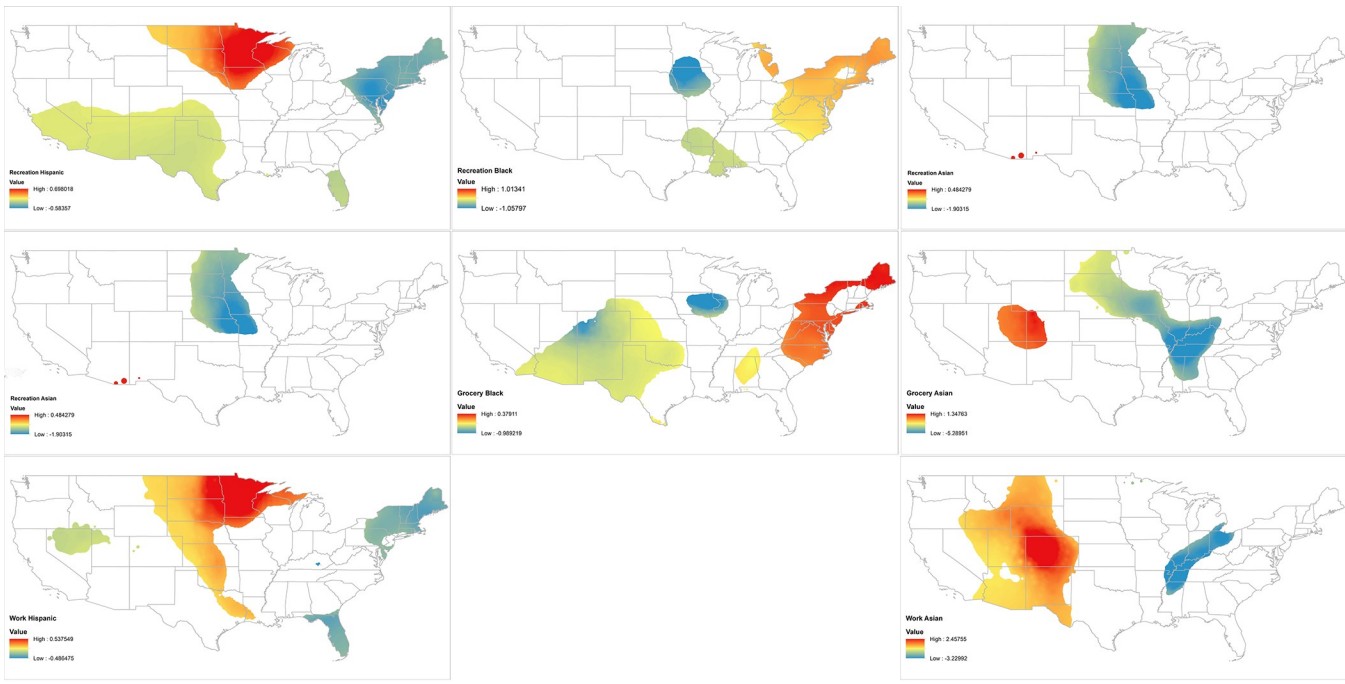

**Fig 3. Select GWR coefficients.** Source TIGER/line. Note: Nonsignificant (P>0.05) marked in white.

The GWR coefficients range dramatically, suggesting that the relationship between race/ethnicity and distancing depends on the location of a county. For example, the Black coefficients for the recreation model range from -1.061 to 1.015, meaning there are counties that the percent Black can have a positive or negative association with recreation distancing. Even a state that has shelter at home policies can be associated with a range of outcomes, from more distancing to less distancing. These findings raise two critical questions: Which of these coefficients are significant and where, exactly, do they vary? To better contextualize our GWR estimates, we make use of a series of maps of the 48 contiguous states, presented in Fig 3, to visually assess the local spatial relations of social capital to distancing. As above, red areas mean less overall distancing while blue areas mean more distancing. White areas have statistically insignificant coefficients (p-value < 0.05; Gibbons and Schiaffino 2016).

These maps reveal several interesting spatial trends in the racial/ethnic coefficients. For example, while the Hispanic coefficients in the HLM model were fully significant and related to more recreation and grocery distancing, Fig 3 shows these coefficients are only significant in select parts of the United States. The spatial pattern of the Hispanic coefficients is similar in both models–they are related to more recreation and grocery distancing in parts of the Northeast, the South around Florida, and the Southwest. In addition, there are parts of the Midwest, especially around Minnesota and Wisconsin, where the share of Hispanic residents is significantly related to less grocery and recreation distancing. In other words, there are parts of the country where the GWR analysis showed the association of percent Hispanic has a different relationship to grocery and recreation distancing than was reported in the HLM model. For the Black coefficients, while there are no significant local coefficients for workplace, significant local coefficients can be found for Blacks in the recreational and grocery models. Like the Hispanic models, there are places with significant coefficients of more distancing as well as places with coefficients indicating less distancing. Interestingly, there are numerous places where the Black and Hispanic coefficients overlap but the direction of these relationships diverge. Most

notably, while the percent Hispanic is related to more grocery and recreation distancing in the Northeast, the percent Black is related to less grocery and recreation distancing in these areas. For the Asian populations there are less than clear patterns. While Asian coefficients were significant for less workplace distancing in the HLM models, this is only true in select parts of the United States, including segments of the Midwest.

We compare the findings of Fig 3 to the distribution of distancing reported in Fig 1. There does not appear to be as a strong relationship with distancing as for where the Black and Hispanic coefficients are present. For example, some of the strongest coefficients for Hispanics and work distancing can be found in New England, where the workplace distancing values overall are weak. Alternatively, there is a strong association with Hispanics and less grocery distancing in the Midwest, a place that has rather strong grocery distancing values, overall.

Next, we compare our results in Fig 3 to the distribution of non-Whites and their corresponding spatial isolation index scores reported in Fig 2. Reflecting the HLM models, this visual assessment identifies only minimal support of the relationship of COVID-19 distancing to segregation. We start with the Black/White isolation index for recreation, which, recall, was the only significant segregation coefficient in the HLM model. There is not a clear correlation with respect to the location of high Black/White isolation and COVID-19 distancing. By contrast, there are places where high isolation values overlap with coefficients in the Northeast, indicating less distancing in that area. For Hispanic populations, meanwhile, there are high Hispanic/White isolation scores in the Southwest and Florida that loosely correspond to Hispanic recreation and grocery distancing coefficients.

## Discussion

This study explores how racial/ethnic segregation in a particular area predict physical distancing in response to COVID-19 across the United States. Our results contribute to our understanding of race/ethnicity and pandemic response by demonstrating the local spatial nuance of this relation. While research covering previous pandemics and disasters suggest Black and Hispanic populations would uniformly be less likely to distance than White populations, our results paint a more varied picture: across the 48 contiguous states, there are places where the share of Black and Hispanic groups can be related to either more distancing or less distancing. Given these findings, we only have mixed support for our first hypothesis, *H1. Regardless of the type of COVID-19 physical distancing, racial/ethnic minorities will be less likely to distance than Whites*.

Support for our second hypothesis, *H2. Regardless of the type of COVID-19 physical distancing, Black populations will be less likely to distance than Hispanics*, is also mixed. On the one hand, the HLM model shows Hispanic populations are more able to distance from recreation and grocery sites than Black populations. The GWR model builds on this, identifying locations in the Northeast where percent Hispanic was related to more recreation/grocery distancing while percent Black was related to less. This model also identified areas where percent Hispanic related to more workplace distancing and percentage of Blacks did not. On the other hand, the GWR model also shows there are places where only the percent Black is related to more recreation and grocery distancing. As such, while Hispanic populations appear to have some advantage in physical distancing that Black populations do not, this is not unilateral across the United States.

We can only explain the local variation of Black and Hispanic results to a certain degree. As we control for forces that shape the ability of people to distance, like poverty and unemployment, the race/ethnicity coefficients point to local variation in enabling factors associated with COVID-19 distancing [10, 71]. *How* exactly race/ethnicity affect the willingness of populations

to distance is harder to pinpoint. None of the theories concerning race/ethnicity and health behaviors discussed in the literature review adequately explain the spatial variability of local coefficients. For example, we initially suspected disadvantage eroding social support for the Black community could explain the differences in recreation/grocery distancing for Black and Hispanic populations in the Northeast. However, this does not explain the difference in coefficients in parts of the Midwest, where Hispanics are less likely to distance and the Black population is more likely to distance. Also, segregation appears related to some distancing for Hispanic populations, but it too is a limited explanation.

We suspect our model is capturing some localized characteristics of race/ethnicity that have been identified by others. For one, the character of discrimination has been found to vary across regions [82]. As Baldwin [83] once observed, "I tried to explain what has happened, unfailingly, whenever a significant body of Negroes move North. They do not escape jim crow: they merely encounter another, not-less-deadly variety." Consequentially, the influence of discrimination and its consequences on distancing in response to COVID-19 may be different from region to region. Alternatively, the social support networks of non-White communities have also been found to vary at a granular level within and between where their populations are high [84]. Put differently, the effectiveness of ethnic bonds may not be uniformly useful even in the presence of high ethnic concentrations. We do not have sufficient data to substantiate either of these, or other, possibilities to explain the varying influence of race/ethnicity. Nonetheless we can conclude that due to local factors, Black and Hispanic populations do not consistently experience the same challenges in distancing. Each group has unique advantages and disadvantages dependent on local context.

Another important consideration in evaluating the relationship of race/ethnicity to distancing is the site from which they are avoiding. The local racial/ethnic coefficients for the different COVID-19 distancing sites did not always correspond. In particular, the Black and Hispanic populations did not distance from workplaces in the same way they had from recreational sites or grocery stores. Some of these differences could be due to the kinds of localized disadvantage by race/ethnicity discussed above. The presence of less workplace distancing for Hispanics in the Midwest and lack of workplace distancing coefficients in the Southwest could be pointing to the heavy presence of agriculture in these areas, which disproportionately impacts Hispanic populations. We do not have sufficient data to make conclusive arguments as for why these differences exist.

We also addressed how residential segregation relates to COVID-19 distancing for non-White populations. We did not have sufficient evidence from our models to support our third hypothesis, *H3. Higher racial/ethnic segregation relates to more COVID-19 physical distancing for racial/ethnic minorities*. In our HLM models we found evidence that White populations were less likely to distance in White/Black segregated places but did not find a measurable advantage for Black populations in these places. We did observe in the GWR models an overlap between places with high Hispanic spatial isolation and recreation and grocery distancing, namely in the Southwest and to a lesser degree the South and Northeast. Another dimension of segregation, like evenness, may yield different results than what we found. However, this would capture effects other than social environment [44]. Also, we were unable to directly measure the segregation within the COVID-19 distancing sites themselves: workplaces, grocery stores, and recreational sites. A study of this segregation could suggest more direct relationships than we were able to identify through residential segregation [9].

Our results offered clear support of our fourth hypothesis, *H4. The association of race/ethnicity to COVID-19 physical distancing will not be consistent across the United States*. As well as shedding light onto the subtle ways in which racial/ethnic populations relate to COVID-19 distancing, our results provide an effective demonstration of the value of GWR. Our HLM models

identified only minimal relationships between race/ethnicity and physical distancing while the GWR models revealed a varied and in-depth association of race/ethnicity to distancing across the United States. These results make clear the importance of not presuming the coefficients of conventional regression models carry equally across the U.S. Spatial methods like GWR are a desirable way to curb these limitations.

To summarize, race and ethnicity matter for population-level COVID-19 physical distancing. Policy makers should continue efforts to ensure people of color are able to effectively distance. However, we cannot assume non-White groups will respond monolithically to COVID-19. In finding ways to ensure people of color can distance, it is pivotal to account for local variations that may inhibit or facilitate these efforts. GWR is a promising tool to identify these variations. We can learn from the places that effectively distanced to inform policy interventions in the places which were less successful at distancing. COVID-19 is not the last time we will face this kind of crisis. It is imperative that we have the best information possible moving forward to meet the challenges they will present.

## Acknowledgments

### Declarations

Ethical approval: This article does not contain any studies with human participants or animals performed by any of the authors.

Informed consent: Informed consent was obtained from all individual participants included in the study.

## Author Contributions

**Conceptualization:** Joseph Gibbons.

**Data curation:** Joseph Gibbons.

**Formal analysis:** Joseph Gibbons.

**Funding acquisition:** Joseph Gibbons.

**Investigation:** Joseph Gibbons.

**Methodology:** Joseph Gibbons.

**Project administration:** Joseph Gibbons.

**Resources:** Joseph Gibbons.

**Software:** Joseph Gibbons.

**Supervision:** Joseph Gibbons.

**Validation:** Joseph Gibbons.

**Visualization:** Joseph Gibbons.

**Writing – original draft:** Joseph Gibbons.

**Writing – review & editing:** Joseph Gibbons.

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
