## [Decision Letter · Decision Letter 0]

7 Apr 2021

PONE-D-21-07358

Distancing the Socially Distanced: Racial/Ethnic Composition’s Association with Physical Distancing in Response to COVID-19 in the U.S.

PLOS ONE

Dear Dr. Gibbons,

Thank you for submitting your manuscript to PLOS ONE. After careful consideration, we feel that it has merit but does not fully meet PLOS ONE’s publication criteria as it currently stands. Therefore, we invite you to submit a revised version of the manuscript that addresses the points raised during the review process.

We look forward to receiving your revised manuscript.

Kind regards,

Xiaozhao Yousef Yang, Ph.D.

Academic Editor

PLOS ONE

Journal Requirements:

'The funders had no role in study design, data collection and analysis, decision to publish, or preparation of the manuscript.'

4. We note that Figures 1-3 in your submission contain map images which may be copyrighted.

a. You may seek permission from the original copyright holder of Figures 1-3 to publish the content specifically under the CC BY 4.0 license. 

Additional Editor Comments:

I suggest revising the abstract as it currently narrates the findings in a very descriptive manner. Inserting findings with numericals and highlight the key findings instead of "for examples" may be better.

Please explain in greater details why only three types of loci were chosen and what are the theoretical and technical basis of not testing other types of loci. What about the robustness test and sensitivity of using other location in proxy of the current ones?

There is an underdeveloped between racial composition's heterogeneity or segregation and spatial heterogeneity. One is a spatial measure of differentiation, the other is a social construct. More nuanced linkages between the two concepts could be helpful.

Reviewers' comments:

Reviewer's Responses to Questions

**Comments to the Author**

1. Is the manuscript technically sound, and do the data support the conclusions?

Reviewer #1: Yes

2. Has the statistical analysis been performed appropriately and rigorously? 

Reviewer #1: Yes

3. Have the authors made all data underlying the findings in their manuscript fully available?

Reviewer #1: Yes

4. Is the manuscript presented in an intelligible fashion and written in standard English?

Reviewer #1: Yes

5. Review Comments to the Author

Reviewer #1: Heat maps are very well done. Social capital has large amounts of literature and research potential that may be extended upon. Some sources have differing definitions of geography i.e. "Midwest" in Logan JR is different than the NYT. However, final results mitigate the importance of this differentiation.

6. PLOS authors have the option to publish the peer review history of their article (what does this mean?). If published, this will include your full peer review and any attached files.

Reviewer #1: No

---

## [Author Response · Author response to Decision Letter 0]

20 Apr 2021

#Editor Comments#

#I suggest revising the abstract as it currently narrates the findings in a very descriptive manner. Inserting findings with numericals and highlight the key findings instead of "for examples" may be better.#

I have added to the abstract more discussion of the key findings. This addition necessitated additional description given the context required to correctly interpret these coefficients. Nonetheless, I kept this section in line with the PLOS One submission guideline to “Summarize the most important results and their significance.”

#Please explain in greater details why only three types of loci were chosen and what are the theoretical and technical basis of not testing other types of loci.#

These three sites were chosen because they had the least amount of missing data, the other locations available from Google had far fewer available counties with data.

#What about the robustness test and sensitivity of using other location in proxy of the current ones?#

I understand your concern and I have attempted to understand if there is any systematic bias in the Google data. A recent study (Barrios et al. 2021) reported that while the Google mobility data coverage at the county level is relatively limited, the measures are fairly robust and yield findings comparable with other data sources (e.g., Unacast). As such, I argue the Google data show a certain level of external validity and acknowledge this issue as a limitation in the revised manuscript. 

#There is an underdeveloped between racial composition's heterogeneity or segregation and spatial heterogeneity. One is a spatial measure of differentiation, the other is a social construct. More nuanced linkages between the two concepts could be helpful.#

I expanded the discussion of my racial composition and isolation measures in the methods section, highlighting how they related to one another.

#Reviewer Comments#

#Heat maps are very well done.# 

Thank you very much.

#Social capital has large amounts of literature and research potential that may be extended upon.# 

We have added a new paragraph to the literature review to discuss social capital and its potential role in distancing.

#Some sources have differing definitions of geography i.e. "Midwest" in Logan JR is different than the NYT. However, final results mitigate the importance of this differentiation.#

We have dropped the sentence in the conclusion which equates the NYT definition of Midwest to the Census-designated version used in our analysis.

---

## [Editor Report · Decision Letter 1]

7 May 2021

Distancing the Socially Distanced: Racial/Ethnic Composition’s Association with Physical Distancing in Response to COVID-19 in the U.S.

PONE-D-21-07358R1

Dear Dr. Gibbons,

We’re pleased to inform you that your manuscript has been judged scientifically suitable for publication and will be formally accepted for publication once it meets all outstanding technical requirements.

Kind regards,

Xiaozhao Yousef Yang, Ph.D.

Academic Editor

PLOS ONE
---

## [Editor Report · Acceptance letter]

17 May 2021

PONE-D-21-07358R1 

Distancing the Socially Distanced: Racial/Ethnic Composition’s Association with Physical Distancing in Response to COVID-19 in the U.S. 

Dear Dr. Gibbons:

I'm pleased to inform you that your manuscript has been deemed suitable for publication in PLOS ONE. Congratulations! Your manuscript is now with our production department. 

Kind regards, 

on behalf of

Dr. Xiaozhao Yousef Yang 

Academic Editor

PLOS ONE